# The Local Neuropeptide System of Keratinocytes

**DOI:** 10.3390/biomedicines9121854

**Published:** 2021-12-07

**Authors:** Nicola Cirillo

**Affiliations:** Melbourne Dental School, University of Melbourne, 720 Swanson Street, Carlton, Melbourne, VIC 3053, Australia; nicola.cirillo@unimelb.edu.au

**Keywords:** keratinocytes, neuropeptides, hypothalamic-pituitary-adrenal axis, hypothalamic–pituitary–thyroid axis, neuroendocrine system

## Abstract

Neuropeptides have been known for over 50 years as chemical signals in the brain. However, it is now well established that the synthesis of this class of peptides is not restricted to neurons. For example, human skin not only expresses several functional receptors for neuropeptides but, also, can serve as a local source of neuroactive molecules such as corticotropin-releasing hormone, melanocortins, and β-endorphin. In contrast, an equivalent of the hypothalamic-pituitary axis in the oral mucosa has not been well characterized to date. In view of the differences in the morphology and function of oral mucosal and skin cells, in this review I surveyed the existing evidence for a local synthesis of hypothalamic-pituitary, opiate, neurohypophyseal, and neuroendocrine neuropeptides in both epidermal and oral keratinocytes.

## 1. Introduction

The term neuropeptide was originally coined to indicate small protein molecules that are contained in neurons [1]. They act either in an endocrine manner, where they reach their target cells via the bloodstream or a paracrine manner, as co-transmitters modulating the function of neurotransmitters [2]. Although by definition the biosynthesis of neuropeptides must occur in neurons, it is important to keep in mind that neuropeptides are not just in the brain—they act both in and out of the central nervous system [3]. Furthermore, it is now recognized that this class of molecules is expressed in many other tissues, including skin-resident cells [2]. The skin is considered an important peripheral neuro-endocrine-immune organ that is tightly networked to central regulatory systems. Specifically, epidermal and dermal cells produce and respond to classical stress neurotransmitters, neuropeptides, and hormones [4].

Keratinocytes cover both the skin and oral mucosa, but the morphology of these tissues and the behavior of the keratinocytes from these two sites are different [5]. One significant dissimilarity between the two sites is the response to injury, where oral mucosal wounds heal faster and with less inflammation than equivalent cutaneous wounds [5]. Unlike skin and its appendages, a mucosal equivalent of the neuroendocrine system has not been characterized. Here, I examine the evidence demonstrating the response to, and biosynthesis of, neuropeptides in the epithelial cells lining the skin and oral mucosa (Table 1).

## 2. Hypothalamic Releasing Factors

### 2.1. Corticotropin Releasing Hormone (CRH)

CRH is chemically classified as a neuropeptide hormone and acts as central coordinator for neuroendocrine and behavioral responses to stress with the activation of the hypothalamic-pituitary-adrenal (HPA) axis. The process begins with the hypothalamic release of corticotropin releasing hormone (CRH) which stimulates the production of adrenocorticotropic hormone (ACTH) and other proopiomelanocortin (POMC) peptides via the activation of CRH receptor type 1 (CRH-R1) [6]. ACTH in turn acts on the adrenal cortex, which produces glucocorticoid hormones.

The skin has an equivalent of the central HPA axis and therefore is capable of producing CRH and of activating the downstream biosynthetic cascades [7]. Keratinocytes are also responsive to CRH stimulation by expressing functional CRH receptors (CRH-R) that respond to CRH and urocortin peptides (exogenous or produced locally) through activation of receptor(s)-mediated pathways to modify skin cell phenotype [7]. Although similar to its systemic equivalent, the cutaneous HPA axis is particularly responsive to local stressors (e.g., solar, thermal, chemical, biological insults). For example, UV light and bacterial extracts induce CRH synthesis and hence activate the neuronal, endocrine and immune systems in the skin [8,9]. Surprisingly, a similar local system of CRH production has not been demonstrated in the resident cells of the oral epithelium.

### 2.2. Growth Hormone-Releasing Hormone (GHRH)

The principal action of growth hormone-releasing hormone (GHRH) is considered to be the stimulation of GH production and release in the pituitary. GHRH also induces the proliferation of pituitary cells by binding to the pituitary type GHRH receptor (pGHRH-R) [10]. In addition to the hypothalamus, expression of GHRH has been reported in several nonhypothalamic tissues including placenta, ovaries, testes, lymphocytes, and others [11,12].

Much evidence suggests that besides its GH-stimulating action on the pituitary gland, GHRH plays a role in extrapituitary tissues. For example, GHRH accelerates wound healing and tissue repair by acting primarily on wound-associated fibroblasts [12]. Normal human fibroblasts respond to agonistic analogs of GHRH by increased proliferation, but this activity is abolished in senescent cells.

The extrapituitary effects of GHRH in peripheral tissues, including cancers, are mediated, at least in part, by the splice variant of GHRH receptor (SV1). In malignant cells, GHRH stimulates tumor growth by paracrine and autocrine mechanisms [13]. The stimulatory loop, formed by tumor-derived GHRH and its receptors, can be blocked by GHRH antagonists, resulting in inhibition of tumor growth in experimental models [12,13,14].

With regards to the oral mucosa, the expression of GHRH splice variant 1 receptor has been reported in neoplastic lesions of the oral cavity. In one study [14], anti-SV1 immunoreactivity was detected in only 9% (three of 33) precancerous lesions (one hyperplasia and two dysplasias), while 44% (12 of 27) carcinomas were positive for SV1 (*p* < 0.002).

### 2.3. Somatostatin (SST)

Somatostatin (SST, also abbreviated as SOM or SS) or GH-inhibiting hormone, was originally isolated from hypothalamus and has later been detected in other brain regions, the peripheral nervous systems and in the gastrointestinal tract [15].

In the skin, immunoreactivity to SST-14—the predominant form of SST in the brain—has been demonstrated by confocal laser scanning microscopy in Merkel cells, in the vicinity of sweat glands, in keratinocytes, in Langerhans cells (LC) and in suprabasal cells of the epidermis [16]. SST can be immunostained in hair follicle cells, especially in the outer sheath, and cultured sheath cells do secrete this neuropeptide [17]. In skin keratinocytes, SST controls a number of physiological processes including cell adhesion, permeability, and wound healing.

Although there is no direct evidence for the production of SST in the oral mucosa, research has provided insights into the expression and regulation of SST receptors in cells and tissues of the periodontium. In one example, it was shown that proinflammatory, microbial and obesity-associated molecules result in an SSTR2 upregulation in human periodontal ligament fibroblasts. This signifies that SST is likely to have an effect in the tissues of the oral cavity [18].

### 2.4. Thyrotropin Releasing Hormone (TRH)

TRH represents the upstream member of the hypothalamic-ituitary-thyroid (HPT) axis and controls thyroid hormone production through regulating the release of thyroid-stimulating hormone (TSH) in the anterior pituitary gland [19]. In addition to the hypothalamus, TRH is expressed in several nonhypothalamic tissues including the heart, gastrointestinal tract, and reproductive organs [20].

The expression of HPT axis-related genes has been detected in normal human skin [21,22]. This includes TRH mRNA in cultured human epidermal keratinocytes, dermal fibroblasts, and hair follicle (HF) papilla fibroblasts [21]. In addition to being synthesized locally by skin cells, TRH controls local TSH production as demonstrated by increased intraepidermal TSH immunoreactivity upon incubation with exogenous TRH [22]. Thus, human epidermis can be considered both an extra-pituitary source and target of the HPT axis. These findings currently do not extend to the oral mucosa where the local synthesis of TRH and response to TRH stimulation has not been convincingly demonstrated so far.

## 3. Pituitary Hormones

### 3.1. Adrenocorticotropic Hormone (ACTH)

It has recently become apparent that production of ACTH (corticotropin) is not restricted to the pituitary. Among other cell types, it is synthesized and released by human keratinocytes in response to a number of stimuli including phorbol myristate acetate, ultraviolet light and interleukin-1 [23]. For example, ultraviolet B (UVB) radiation stimulates increased expression of the proopiomelanocortin (*POMC*) gene which is accompanied by production and release of α-melanocyte stimulating hormone (α-MSH) and adrenocorticotropin (ACTH) by both normal and malignant human melanocytes and keratinocytes [24].

Research suggests that corticotropin may have an immunoregulatory role in oral mucosa. Studies have investigated ACTH effects on a human oral keratinocyte cell lines and shown that corticotropin, acting via its specific receptor, stimulates a dose-dependent increase in DNA synthesis and induces cell proliferation, thus identifying corticotropin as a mitogenic regulatory peptide of keratinocytes [25]. While the evidence for a local synthesis of ACTH in the oral cavity is debated, it is well known that the oral mucosa can respond to ACTH stimulation. For example, high levels of circulating ACTH results in oral hyperpigmentation via stimulation of the α-MSH receptors in melanocytes [26]. A point mutation in the ACTH receptor resulting in high plasma ACTH levels manifests with dark coloration of the oral mucosa and gums [27]. Corticotropin also exerts direct effects in oral mucosal cells [28]. These observations are likely related to the fact that oral fibroblasts and keratinocytes express ACTH receptor (MC2R) and can activate its downstream signaling, e.g., de novo synthesis of cortisol [28].

Activation of corticotropin-mediated biosynthetic pathways is functioning in oral tumours and may have clinical implications. For example, ACTH can reduce the effectiveness of chemotherapeutic agents such as doxorubicin in oral squamous cell carcinoma (OSCC) cell lines, possibly via the autocrine effect of cortisol [29]. Cancer-derived cortisol induces a glucocorticoid receptor (GR)-dependent inhibition of tumour-specific CD8+ T cells [30] and, consistent with this observation, higher levels of ACTH in the microenvironment of OSCCs are associated to a reduced density of the lymphoplasmacytic infiltrate [31].

Thus it is clear this polypeptide has remarkable effects in the physiological and pathological processes of the oral mucosa.

### 3.2. α-. Melanocyte Stimulating Hormone (α-MSH)

The melanocyte-stimulating hormones (MSHs) derive from the precursor hormone, proopiomelanocortin (POMC), and α-melanocyte-stimulating hormone (α-MSH) binds the melanocortin 1 receptor (MC1R) with the highest affinity [32]. Activation of the downstream effectors of MC1R signaling results in an increase of proliferation and melanogenesis of human melanocytes [32].

Growing evidence indicates that MC1R and its ligand α-MSH have other functions in the skin in addition to pigment production in melanocytes. Activation of the MC1R/α-MSH signaling pathway has been implicated in the regulation of both inflammation and extracellular matrix homeostasis. In particular, there is a pattern of spatial and temporal localization of these molecules during cutaneous wound repair [33]. In one study, MC1R and α-MSH protein levels were upregulated in human burn wounds and hypertrophic scars compared to uninjured human skin, where receptor and ligand were absent [34]. In burn wounds and hypertrophic scars, MC1R and α-MSH localize to epidermal keratinocytes and dermal fibroblasts [34]. The local synthesis of α-MSH has been confirmed recently with the demonstration that keratinocytes in UV-irradiated skin produce and secrete α-MSH [35].

In the oral mucosa, it has been shown that human oral epithelial cells and fibroblasts respond to α-MSH stimulation in vitro [36]. Furthermore, α-MSH has potential effects in promoting human pulp fibroblast adhesion and cell proliferation and can also reduce the inflammatory state of LPS-stimulated pulp fibroblasts. In particular, when delivered via poly-caprolactone (PCL) membranes, it induced proliferation of pulp fibroblasts, whereas free α-MSH inhibited this proliferation [37].

### 3.3. Growth Hormone (GH)

The growth hormone (GH) via its receptor mediates a wide range of growth-related and metabolic actions, both directly and via insulin-like growth factor 1 (IGF-1) [38].

GH receptors are found in almost all cell types forming the skin, while IGF-1 receptors’ expression is restricted to the epidermal keratinocytes [39]. Both GH excess, as in case of acromegaly in adults or gigantism in growing children, and GH deficiency states lead to skin manifestations. In case of GH excess the main dermatological findings are skin thickening, coarsening of facial features, acrochordons, puffy hands and feet, oily skin and hyperhidrosis, while GH deficiency, on the contrary, is characterized by thin, dry skin and disorder of normal sweating [39]. Mechanistically, GH enhances the local formation of IGF-1, which activates fibroblast proliferation and keratinocyte migration [40].

Interestingly, different research groups have transduced human and mouse keratinocytes with the *GH* gene with a view of using the epidermis as a vehicle for ex vivo gene transfer and systemic delivery [41,42]. This suggests that keratinocytes, including oral keratinocytes, are not a primary site of GH synthesis.

### 3.4. Prolactin (PRL)

The principal role of prolactin (PRL) in mammals is the regulation of lactation [43]. However, new roles of prolactin in human health and disease have recently been discovered, particularly its involvement in metabolic homeostasis including body weight control, adipose tissue, skin and hair follicles, pancreas, bone, and the adrenal response to stress [43]. It is now recognized that human skin and its appendages locally express this pleiotropic neurohormone that regulates hair follicle cycling, angiogenesis, keratinocyte proliferation, and epithelial stem cell functions [44]. Two key endocrine controls of pituitary PRL secretion, oestrogen and TRH, have been reported to regulate PRL and PRLR expression in human skin [45]. In contrast, other studies found that while the PRL receptor was upregulated after culture confluence, in differentiating keratinocytes, the authors were unable to detect any cellular response to PRL [46].

Although the biosynthesis of PRL in the oral cavity has not been demonstrated so far, the presence of hPRLR in human periodontal ligament (PDL) fibroblasts together with PRL-induced upregulation of osteogenic markers strongly suggest a direct regulatory role of PRL in PDL and periodontal tissue development [47].

### 3.5. Thyroid Stimulating Hormone (TSH)

TSH or thyrotropin is a classical hormone secreted by the anterior pituitary gland that controls thyroid hormone production. However, published evidence suggests that TSH is also expressed in peripheral tissues such as epidermal keratinocytes and dermal fibroblasts and that keratinocytes express functional receptors for thyroid-stimulating hormone [48,49]. Specifically, Western blot and immunohistochemical analyses of skin specimens confirmed the presence of TSH-R protein in keratinocytes and fibroblasts. Moreover, TSH treatment induced the proliferation of cultured keratinocytes and fibroblasts and increased keratinocyte intracellular cAMP [50]. The TSH receptor detected in the skin has shown to be functionally active, which could explain the pathogenesis of skin lesions in the course of Graves’ disease [51].

Selective expression of the gene for TSH-β was found in keratinocytes [21], however it is not entirely clear whether TSH production can be regulated autocrinally or paracrinally by local TRH. For example, some studies have failed to demonstrate the expression of TRH receptors or TSH ligand in the epidermis in situ [21,22]. Hence, the function and implications of TSH biosynthesis in oral and skin keratinocytes has not yet been completely elucidated. In particular, there seems to be substantial differences between in vitro and in vivo synthesis of members of the HPT axis. Overall, these findings suggest that TSH expression is not constitutive in keratinocytes, but instead might respond to environmental stimulation.

## 4. Opiate Peptides

The endogenous opioid system consists of 3 families of neuropeptides, β-endorphin, enkephalins, and dynorphins, and 3 families of receptors, μ (MOR), δ (λ, DOR), and κ (KOR) [52], which are widely distributed in the central and peripheral nervous system and gastrointestinal tract. The best known effects of opioids include analgesia, sedation, respiratory depression, and constipation [53].

There is growing evidence that opioid receptors and their endogenous opioid agonists are functional in different skin structures. Specifically, research has shown that stimulating effects exist of endogenous and exogenous opioids on the migration, formation of granulation, and re-epithelialization in keratinocytes [54]. Early studies in skin keratinocytes showed that enkephalins (methionine-enkephalin, leucine-enkephalin) inhibited cell differentiation dose-dependently, while beta-endorphin had no effect [55]. β-noendorphin, an opioid peptide derived from the proteolytic cleavage of prodynorphin, accelerates wound repair in human keratinocytes through the increase of keratinocyte migration without affecting cell proliferation [56].

In addition to responding to opiate peptides, it has been shown that keratinocytes are a primary source of opioids. In one study, analysis of microarray data demonstrated that cultured keratinocytes had a functional neuroendocrine machinery, and this was confirmed by testing the secretion of six neuroactive molecules by ELISA, namely α-MSH, β-endorphins, melatonin, substance P, cortisol, and neurotensin. Interestingly, hyaluronic acid regulated the production of several neuropeptides, including β-endorphins, in vitro [57]. Pro-enkephalins and met-enkephalin are also expressed in skin keratinocytes in vivo [58]. In general, significant expression of enkephalins occurs in fibroblasts and keratinocytes, including Leu- and Met-enkephalin. Proenkephalin gene expression in keratinocytes increases in a time- and dose-dependent manner in response to UVR, Toll-like receptor TLR4, and TLR2 agonists and is altered in pathological skin conditions [59].

Oral mucosal cells also respond to opioid stimulation and are likely a source of local production of these neuropeptides. For example, the application of morphine in oral epithelial keratinocytes enhances cell migration and wound closure through δ-opioid receptors. The expression of the opioid receptors MOR, DOR and KOR on primary human oral epithelial cells has also been verified [60]. Both opioid growth factor (OGF), or [Met-5]-enkephalin, and OGF receptor were colocalized in the paranuclear cytoplasm and in the nuclei of keratinocytes in the stratum basale [61]. Importantly, these analgesic neuropeptides are produced in epithelial tumours of the mouth. In one study, oral squamous cell carcinoma (SCC) derived from human tongue SCC expressed β-endorphin, leu-enkephalin and dynorphin [62]. Hence, oral keratinocytes are an active site of local production of opioids and may be involved in pain control in physiological and pathological processes of the oral mucosa.

## 5. Neurohypophyseal Peptides

Oxytocin (OXT) mediates a wide spectrum of tissue-specific actions, ranging from cell growth, cell differentiation, sodium excretion to stress responses, reproduction and complex social behavior. A large body of evidence now supports the role for oxytocin in skin physiology and aging [63], whereas the role of vasopressin, the other neurohypophyseal peptide, has not been investigated extensively in skin cells.

RT-PCR studies have confirmed the expression of oxytocin in both skin and cultured epidermal keratinocytes. OXT mRNA is transcribed in human epidermal keratinocytes and is released in response to calcium influx via P2X receptors [64]. OXT and its receptor are also expressed in primary human dermal fibroblasts as well as keratinocytes. The OXT system modulates key processes which are dysregulated in atopic dermatitis such as proliferation, inflammation and oxidative stress responses [65].

There is limited evidence for the local production of vasopressin in skin cells. In one study using a novel microarray to evaluate stress-related genes in skin, transcription of vasopressin precursor genes was found to be upregulated in ultraviolet irradiated human epidermal keratinocytes [66].

To the best of our knowledge, no evidence exists to date of the production of these two neuropeptides in oral keratinocytes.

## 6. Neuroendocrine Peptides

This class of neuropeptides includes atrial natriuretic peptide (ANP), vasoactive intestinal peptide (VIP), and calcitonin gene-related peptide (CGRP).

VIP has been shown to affect the proliferative activity of human keratinocytes in vitro [67]. The findings that topical treatments of grafted bioengineered human skin with the antimitotic agent colchicine select for keratinocyte progenitors that express ANP in mice suggest that basal production of ANP in normal skin is negligible [68]. With regards to the oral mucosa, liposomal VIP potentiates DNA synthesis in cultured hamster oral keratinocytes, which suggests a possible response to this peptide [69].

Calcitonin gene-related peptide (CGRP) is a vasodilatory peptide that has been detected at high levels in the blood and cerebrospinal fluid (CSF) under a variety of inflammatory and chronic pain conditions [70]. CGRP has roles in regulating the function of components of the immune system including T cells, B cells, dendritic cells, endothelial cells, and mast cells, and mediates inflammatory and vascular effects in the skin [71].

Indirect evidence of an effect of CGRP on skin physiology was provided by a report of impaired wound healing in a migraine patient undergoing calcitonin gene-related peptide receptor antibody treatment [72]. In one study, a keratinocyte cell line was used to identify the presence of substance P (SP) and CGRP receptors and demonstrated the effects of SP and CGRP stimulation on keratinocyte neuropeptide signaling, cell proliferation, and interleukin-1β (IL-1β), interleukin-6 (IL-6), tumor necrosis factor α (TNF-α), and nerve growth factor (NGF) expression [73]. CGRP also increases proliferation of non-tumorigenic human HaCaT keratinocytes by activation of MAP kinases [74].

Transcriptome microarray, quantitative Polymerase Chain Reaction (qPCR), and Western blot analyses using laser-captured mouse epidermis from transgenics, monolayer cultures of human and mouse keratinocytes, and multilayer human keratinocyte organotypic cultures, have convincingly demonstrated that keratinocytes locally produce CGRP, predominantly its beta isoform [75]. Mouse skin is also positive to CGRP immunoreactivity where this neuropeptide orchestrates wound repair [76].

While CGRP manifests in the dental pulp of rats during experimental tooth movement [77], it is not entirely clear whether oral keratinocytes synthesise CGRP. To this regard, it is interesting to note that adrenomedullin, a vasoactive peptide that shows homology with the CGRP, has mitogenic effects on human oral keratinocytes [78] and is overexpressed in pathogen-challenged oral epithelial cells [79].

## 7. Conclusions

Keratinocytes represent both a source and target of neuropeptides. An ever increasing number of effects of this peripheral neuropeptidergic system includes modulation of cell proliferation, wound healing and inflammation. The local synthesis of several neuropeptides also takes place in keratinocytes, particularly in the skin. For example, a large body of evidence demonstrates the importance of the HPA axis equivalent in epidermal tissues. In contrast, there is little information as to whether a non-neuronal neuropeptidergic system exists in oral keratinocytes. Although not primarily expressed in oral mucosal cells, most neuropeptides have been shown to exert significant effects in the regulation of the physiology of mucosal cells as well as of epithelial tumours of the mouth. Further research is warranted to address whether oral keratinocytes can be considered as an active site of local production of neuropeptides.

## Figures and Tables

**Table 1 biomedicines-09-01854-t001:** Evidence for a neuropeptidergic system in keratinocytes from skin and oral mucosa.

	Epidermal Keratinocytes		Oral Keratinocytes	
**Neuropeptide family**	Local synthesis	Response to exogenous stimulation	Local synthesis	Response to exogenous stimulation
Hypothalamic releasing factors	CRH, somatostatin, TRH	CRH, (GHRH), TRH somatostatin	-	(GHRH), (somatostatin)
Pituitary hormones	ACTH, α-MSH, prolactin, (TSH)	ACTH, α-MSH, (prolactin), (TSH)	-	ACTH, α-MSH
Opiate peptides	β-endorphin, met/leu-enkephalins, (dynorphin)	met/leu-enkephalins, (β-endorphin), (dynorphin)	β-endorphin, leu-enkephalin, dynorphin	(β-endorphin)
Neurohypophyseal peptides	Oxytocin, (vasopressin)	Oxytocin	-	-
Neuroendocrine peptides	CGRP	VIP, CGRP	-	VIP

CRH, corticotropin releasing hormone; GHRH, Growth hormone-releasing hormone; TRH, Thyrotropin releasing hormone; ACTH, adrenocorticotropic hormone; α-MSH, α-melanocyte stimulating hormone; TSH, Thyroid stimulating hormone; VIP, vasoactive intestinal peptide; CGRP, calcitonin gene-related peptide. Brackets indicate indirect or non-conclusive evidence.

## Data Availability

All data items not included in the manuscript are available upon reasonable request to the corresponding author.

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
