# Peer review of "The Local Neuropeptide System of Keratinocytes"

_biomedicines, 2021, doi:10.3390/biomedicines9121854_

Round 1

Reviewer 1 Report

This review paper covers an interesting phenomenon of neuropeptide action and production in skin cells with a focus on those found in the mouth.  While an interesting topic, the authors does not appear to actually present any direct evidence (aside from the opioid peptides and several cases of oral carcinoma) that any of the neuropeptides presented actually are produced in the mouth.  I think with some rephrasing and some expanding on the sections presented, this review could form a really interesting piece,  but for now it is not ready for publication.

Specific comments:

  1. The title is unusual and rhetorical questions usually are not the manner by which reviews are presented. Instead, perhaps, change it to a definitive statement, something like “Probing local production of neuropeptides by oral keratinocytes.”

However, as the entire review isn’t really about this, but more so is about “non-neuron uses and production of neuropeptides, in relation to keratinocytes and the dermis” I would perhaps work something like that in, instead.

  1. Sections 2.1, 2.2, 2.4 etc. In many cases the author expresses, at the end of each section, that there is no evidence of local oral keratinocyte production.  So, why are these sections in a paper that focuses on oral keratinocyte local production?  I think these sections are great, and maybe just change the scope of the paper, don’t include those negative parts at the end of each section and perhaps, instead, broaden the title and scope of the introduction etc to include all keratinocytes and the implications on oral local production. 
  2. For sections 2.1, 2.2, 2.3 and 2.4 they’re very short and need to have a further discussion. A review paper should expand on the use, what were the main results, and these things are lacking, More detail needs to be included in each of those sections.
  3. Line 113 how is ACTH in the oral cavity “controversial?” Perhaps pick another word or phrase like “widely debated?”
  4. Cut out the first sentence in line 129-130 it is too negative. Instead start with “It is clear this polypeptide…” and keep that sentence to tie-up that section.
  5. Like in the previous comment, remove the negative part at the end of the section with line 149 and begin the sentence with “It has been shown that…” A review is about what has been shown, not about what isn’t shown or what there isn’t evidence for (if there isn’t evidence, do not mention it, just leave it out).
  6. Leave out section 3.3 completely.  Those are compounds later talked about and it’s unprofessional to write “see paragraph 4” in a published review paper.  Instead, simply leave out this section and expand on these in later sections.
  7. Line 167 “In the case…” it should read not “In case of”
  8. Line 175 it says “This suggests…” but what actually suggests this? The author includes a use and a conclusion but no actual results so what implications can be made?
  9. Line 191 What is the “direct regulatory role” that is implied – any thoughts on this?
  10. Line 254 “OXT mRNA is transcribed” is proper sentence structure (not “OXT mRNA is the transcribed”
  11. Remove lines 263-265; I think one of the sentences was kept in from editing by accident (the part about the section not being mandatory, and also there’s no period at the end of the sentence – it seems it was left in by accident). The prior sentence that yet again says “no evidence was found” should just be removed. 
  12. Line 273 – Oral mucosa - is a great addition and should be added to the conclusions (not just the opioids as the only example).

Author Response

Many thanks for your constructive comments. I have amended the manuscript accordingly. 

Reviewer 2 Report

The presented manuscript summarizes information on the expression of neuropeptides in the skin and oral mucosa. The topic is clearly arranged. I have only some minor points:

  1. Keywords are missing.
  2. In paragraph 2.4. the information is unclear. The TRH is mentioned first, then TSH is mentioned without this abbreviation being explained. There is also a lack of information on the function of TSH. This should be added.
  3. Sentence in lines 264 and 265 probably should not be there.

Author Response

Thanks for your comments - these have been addressed in the revised manuscript. 

Round 2

Reviewer 1 Report

Excellent revisions!  It would have been slightly improved if there were a figure that shows either structures or diagrams for some of these compounds (figures always enhance papers, even reviews), however the written work as it stands now is suitable for publication.

Author Response

n/a
